# Pasture vs. Coop: Biomarker Insights into Free-Range and Conventional Broilers

**DOI:** 10.3390/ani14213070

**Published:** 2024-10-24

**Authors:** Constantinos Tellis, Ioannis Sarrigeorgiou, Gerasimina Tsinti, Apostolos Patsias, Evgenia Fotou, Vasiliki Moulasioti, Dimitra Kyriakou, Maria Papadami, Vassilios Moussis, Maria-Eleni Boti, Vasileios Tsiouris, Vassilios Tsikaris, Demokritos Tsoukatos, Peggy Lymberi

**Affiliations:** 1Sector of Organic Chemistry and Biochemistry, Department of Chemistry, University of Ioannina, 45110 Ioannina, Greece; ktellis@uoi.gr (C.T.); e.fotou@uoi.gr (E.F.); v.moulasioti@uoi.gr (V.M.); d.kyriakou@uoi.gr (D.K.); pch01356@uoi.gr (M.P.); vmousis@uoi.gr (V.M.); mempoti82@gmail.com (M.-E.B.); btsikari@uoi.gr (V.T.); dtsoykat@uoi.gr (D.T.); 2Immunology Laboratory, Immunology Department, Hellenic Pasteur Institute, 11521 Athens, Greece; sarijon@pasteur.gr (I.S.); biomintsi@gmail.com (G.T.); 3Microbiology and Chemical Laboratory, Pindos APSI, 45500 Ioannina, Greece; apatsias@pindos-apsi.gr (A.P.); biltsiou@vet.auth.gr (V.T.); 4Unit of Avian Medicine, Faculty of Veterinary Medicine, School of Health Sciences, Aristotle University of Thessaloniki, 54124 Thessaloniki, Greece

**Keywords:** fast-growing broilers, slow-growing broilers, natural antibodies, LPS, CPK, poultry industry, poultry biomarkers

## Abstract

With the rising demand for meat, poultry farming often uses fast-growing chickens in intensive systems. However, free-range systems with slower-growing chickens present a healthier and more sustainable alternative. Our study investigated whether specific blood markers could help differentiate broilers reared in these two systems by examining blood samples from 300 chickens: free-range, slow-growing Sasso broilers, and conventionally raised, fast-growing Ross 308 broilers. The results show that Sasso broilers have higher levels of natural immunity markers (IgM anti-LPS antibodies) and lower levels of tissue-related biochemical markers (creatine phosphokinase: CPK), compared to Ross 308 broilers. By combining these two key markers, we were able to distinguish the two types of broilers with higher accuracy. This method could have important applications in the poultry industry, helping to verify farming methods and improve overall production standards.

## 1. Introduction

The use of pasture-based systems in poultry production is driven by consumers’ demand for specialty “natural” and “sustainability-friendly” meat products [1]. The identification of reliable markers that distinguish rearing conditions in the poultry industry is potentially important for the evaluation of meat quality [2]. Pasture-based systems, such as free-range or organic, have been shown to improve animal welfare compared to conventional coop systems [3,4]. However, beyond husbandry standards applied to industrial-scale poultry, such as free-range and conventional indoor broilers, discriminating biomarkers for rearing systems and meat quality have not yet been established [5].

To this end, it is important to include key information from different parameters to identify effective biomarkers that can act synergistically, increasing their sensitivity and specificity. For example, a fully functional immune system is crucial for good health and survival, which, however, can be threatened by stressors and other adverse environmental stimuli differing between farming conditions [6,7]. On the other hand, meat quality is related to pathophysiological skeletal muscle in broilers dependent on rearing conditions [8]. In some cases, muscle damage can be caused by stressful farming conditions, leading to a further reduction in meat quality and the production of pale, soft, and exudative meat [9,10]. Therefore, immunological and tissue-related parameters in blood, modulated by rearing systems, are ideal candidate biomarkers for distinguishing poultry reared with these systems.

Unlike traditional chicken breeds, which represent naturally reproducing populations with greater genetic diversity, commercial broiler strains like Ross 308 are products of highly controlled breeding programs [11]. These programs target specific traits, such as rapid growth, high meat yield, and feed efficiency. Ross 308 (Aviagen Group, Huntsville, AL, USA), a prime example of such a strain, is optimized for performance in high-density, intensive farming environments. In contrast, slow-growing strains like Sasso broilers (Hendrix Genetics BV, Sabres, France) are renowned for their robustness and adaptability, particularly suited for free-range and organic systems [12]. Their genetic background originates from traditional, heritage breeds, which have been selected for traits such as moderate growth efficiency, disease resistance, and high-quality meat production, particularly under extensive rearing conditions. Sasso broilers are bred to achieve market weight over an extended period, which aligns with higher animal welfare standards by reducing the incidence of health issues commonly seen in fast-growing birds, such as leg deformities and cardiovascular disorders [13].

Selective breeding for growth, as observed in fast-growing strains like Ross 308, may also compromise their immune responsiveness to environmental stimuli in free-range systems [14]. In contrast, indigenous slow-growing breeds like Sasso exhibit superior adaptability to free-range environments, serving as a genetic reservoir for such conditions [15]. Moreover, understanding the physiological and immunological effects of rearing conditions is crucial amid increasing consumer demand for products from alternative systems. Comparing meat quality between slow-growing and fast-growing broilers reveals how farming practices influence outcomes [16]. In a previous study, conventionally raised fast-growing Ross 308 broilers showed better performance and lower induced stress, while free-range slow-growing Sasso broilers demonstrated improved nutritional value and antioxidant profiles [17]. Moreover, the latter showed similar or better meat quality, indicating the potential benefits of alternative farming. Therefore, the identification of immunological and biochemical markers shaping dependently on rearing systems is vital for informed decision-making in sustainable poultry farming. With this in mind, our objective herein was to optimize the utilization of such biomarkers in commercial free-range and conventional rearing systems and genotypes.

## 2. Materials and Methods

### 2.1. Study Population

Chickens were provided by the Agricultural Poultry Cooperative of Ioannina “PINDOS” (Rhodotopi, Ioannina, Greece). All procedures were conducted in accordance with welfare guidelines (ISO 22000-Food Safety Management System) [18]. The study was carried out in the 2019–2022 period in PINDOS poultry farms and 300 broilers were divided into two groups (*n* = 150/group), conventional (C) and free-range (FR), as shown in Table 1. Broiler chickens were placed in commercial poultry farms, which were fully equipped with automatic ventilation, heating, lighting, and feeding systems. Water and feed were offered to all broiler chickens ad libitum, whereas the lighting program and microenvironmental conditions (temperature, humidity, CO_2_, NH_3_) were automatically regulated for all houses according to the current European Union legislation (Council Directive 2007/43/EC) [19].

Blood samples were collected before slaughter and specified according to the manufacturers’ guidelines and specifications. Blood samples were collected by jugular vein puncture into clot activator tubes and centrifuged at 1500 rpm for 10 min at 4 °C. The supernatant (serum) of the samples was collected and stored at −80 °C until use. A range of blood parameters, including IgM natural antibodies (NAbs) targeting self (e.g., actin) and non-self antigens (e.g., lipopolysaccharides, LPS), along with creatine phosphokinase (CPK), total cholesterol, triglycerides, creatinine, aspartate aminotransferase (AST), alanine aminotransferase (ALT), and 8-Isoprostane, were measured to assess their potential to differentiate between the two broiler groups.

### 2.2. Measurements of Immunological Parameters

#### Anti-Actin and Anti-LPS IgM Natural Antibody Levels

Measurements of serum IgM anti-actin and anti-LPS NAbs were conducted by in-house ELISA, as previously described [20]. Briefly, bovine globular actin was prepared as previously described [21] and LPS from Escherichia coli (L2880) was purchased from Sigma Aldrich (St. Louis, MO, USA). High-binding ELISA microplates (Nunc-MaxiSorp, Roskilde, Denmark) were used for antigen coating (10 μg/mL for actin and 5 μg/mL for LPS, in carbonate–bicarbonate buffer at 0.1 M pH 9.6 (CBC). Sera were diluted 1/100 in sample dilution buffer (PBS containing 1% BSA and 0.05% Tween), and alkaline phosphatase-conjugated secondary antibodies against chicken IgM (μ-chain specific) (SAB3700239-1, Sigma Aldrich) was added (1 μg/mL in the sample dilution buffer). Antibody binding was assessed by adding the substrate 4-nitrophenyl-phosphate-disodium salt hexahydrate (pNPP-N2765, Sigma), and the optical density (OD) of the colored product was measured at 405 nm (620 nm reference) in a TECAN photometer (TECAN Spark Control Magellan V2.2, Grödig/Salzburg, Austria).

### 2.3. Measurements of Biochemical Parameters

#### 2.3.1. Creatine Phosphokinase

The determination of CPK levels in serum was assessed spectrophotometrically using the commercial CK liquid NAC activates UV test (reference number 12015 Human Diagnostics worldwide), according to the manufacturer’s instructions, with slight modifications. Briefly, ten (10) μL of serum samples was added to each well of microtiter 96-well plates (Costar UV transparent microplate, Corning Inc., Corning, NY, USA). A calibration curve was simultaneously generated using standard CPK activity (U/L). Following this, the working solution was added with a multichannel pipette to both samples and standards to initiate the reaction. Enzyme reaction in samples was measured with an ELISA reader (Tecan infinite 200 PRO) and the increased rate formation of NADPH at 340 nm using kinetic monitoring (Abs/1 min, for 10 min at 37 °C). The average absorbance differences per minute (ΔA/min) were calculated for samples and standards. The CPK activity of the samples was calculated using a linear standard curve (Appendix A).

#### 2.3.2. CPK Isoenzyme Activity

CPK isoenzyme activity of serum was determined using agarose gel electrophoresis The activities of CK-MM and CM-MB isoforms were determined using the commercially available iso-CK kit (Ref. 4111, Hydragel ISO-CK, Sebia Italia, Florence, Italy) according to the manufacturer’s instructions. Briefly, CK isoenzymes consist of two subunits, M (“muscle”) and B (“brain”), assembled in dimers. The three resulting combinations constitute the three isoenzymes: MM, principally located in cardiac and skeletal muscles, MB in cardiac muscles, and BB in cerebral tissues. Each subunit has a specific electrical charge, which confers characteristic mobility on the individual CK isoenzymes. In the iso-CK kit, plasma samples are electrophoresed and separated CK isoenzymes are visualized using a specific chromogenic substrate as detailed in the manufacturer’s instructions. The resulting gels are ready for visual examination and densitometry to obtain accurate relative quantification of individual bands.

#### 2.3.3. Total Cholesterol

Total cholesterol levels in serum samples were analyzed using the Cholesterol liquicolor Enzymatic kit (reference number 10028; Human Diagnostics worldwide) via spectrophotometric assessment, following the manufacturer’s instructions. Briefly, ten (10) μL of serum was added to microtiter 96-well plates (Costar, Corning Inc., NY), followed by completion to 20 μL with ddH2O. A calibration curve was prepared simultaneously using varying volumes (0, 3, 5, 7, 10, 13, and 15 μL) of the kit’s standard solution (200 mg/dL), adjusting the volume to 20 μL with ddH2O in each standard well. Then, 280 μL of enzyme solution was added to the sample and standard wells with a multichannel pipette to initiate the reaction. After stirring (200 rpm for 1 min), the plates were incubated at 37 °C for 15 min. Color intensity in each well was measured at 500 nm using a microplate reader (Tecan infinite 200 PRO), and total cholesterol levels were calculated using the linear standard curve.

#### 2.3.4. Triglycerides

Triglyceride levels in serum samples were evaluated using the commercial Cholesterol liquicolor Enzymatic kit (reference number 10724 Human Diagnostics worldwide) via spectrophotometric analysis, following the manufacturer’s instructions. Similarly to the cholesterol assay, the kit comprises an enzymatic triglyceride reagent (with substrates, enzymes, and buffer) and a triglyceride standard (200 mg/dL). The procedure mirrors that described for measuring cholesterol.

#### 2.3.5. Creatinine

Creatinine levels in serum samples were determined spectrophotometrically using the Creatinine liquicolor Colorimetric test (reference number 10051 Human Diagnostics worldwide) following the manufacturer’s instructions with slight modifications for microplate use. The assay relies on the Jaffe reaction, where creatinine reacts with alkaline picrate to form a red complex. A specific time interval was selected for measurements to minimize interferences from other serum components. The intensity of the resulting color is directly proportional to the concentration of creatinine in the sample.

#### 2.3.6. Aminotransferases AST and ALT

Aspartate aminotransferase (AST), also known as glutamic oxalacetic transaminase (GOT), levels in serum were measured spectrophotometrically using the GOT (ASAT) IFFC mod. liquiUV test (reference number 12011 Human Diagnostics worldwide), following the manufacturer’s instructions with slight modifications for microplate use. Similarly, alanine aminotransferase (ALT), also known as glutamate pyruvate transaminase (GPT), levels in serum were determined spectrophotometrically using the GPT (ALAT) IFFC mod. liquiUV test (reference number 12012 Human Diagnostics worldwide).

#### 2.3.7. 8-Isoprostanes

The levels of 8-isoprostanes (8-epi-PGF2a) in serum samples were determined using a competitive enzyme immunoassay (commercial 8-isoprostane EIA kit; Cayman Chemicals, Ann Arbor, MI, USA), following the manufacturer’s instructions. Briefly, sera were diluted at 1:50 (*v*/*v*) with EIA buffer as per instructions, and standards were prepared accordingly. In each well of the plate, 50 μL of samples, standards, and controls was added, followed by 50 μL of 8-isoprostane AChE tracer (except for Total Activity; TA, and blank; blk wells) and 50 μL of 8-isoprostane ELISA antiserum (except for TA, Non-Specific Binding; NSB, and blk wells). The plate was covered with plastic film and incubated for 18 h at 4 °C. After incubation, the contents of the wells were discarded, and they were washed 4 times with 300 μL of wash buffer to remove any unbound reagents. Subsequently, 200 μL of Ellman’s Reagent (containing the substrate for AChE) was added to each well, and the enzymatic reaction took place in the dark at room temperature. The yellow color developed within 90–120 min, and the absorbance was measured at 412 nm using an ELISA microplate reader (SpectraMax, Molecular Devices, Sunnyvale, CA, USA). The level of 8-epi-PGF2a in serum samples was calculated using a standard curve. Samples and standards were measured in duplicate.

### 2.4. Statistical Analysis

All statistical calculations were analyzed using SPSS 21.0 software (version 21.0, SPSS Inc., Chicago, IL, USA) and GraphPad Prism version 9.0.0 (GraphPad Software, San Diego, CA, USA). Values with a normal distribution were expressed as mean ± standard deviation, while those without a normal distribution were presented as medians. For two independent group comparisons, the Mann–Whitney U test was used. Correlations were assessed with the non-parametric Spearman test. In all cases, the significance level was set at 5%, tests were two-sided, and a result was considered significant if the estimated *p*-value (*p*) was less than the significance level. Multiple logistic regression analysis with receiver operating characteristic (ROC) curves and their corresponding areas under the curve (AUCs) were used to determine the optimal cut-off values of NAb and CPK levels to differentiate the FR and C groups. The accuracy rate for ROC curves was calculated with a 95% confidence interval (95% CI). Statistical significance was set at *p* < 0.05. All graphs were extracted using GraphPad Prism version 9.0.0.

## 3. Results

### 3.1. Immunological Parameters

Over a period of 24 months, 150 Sasso broilers (FR group) and 150 Ross 308 broilers (C group), raised under free-range and conventional, indoor, industrial-scale farming systems, respectively, were evaluated for a couple of immunological parameters. Levels of serum anti-actin and anti-LPS IgM-NAbs are shown in Figure 1. For IgM NAbs against actin, the mean OD value for Ross 308 broilers was 1.124 ± 0.510, while Sasso broilers, 1.542 ± 0.508 indicated a 1.4-fold higher level (*p* < 0.0001). Similarly, for IgM NAbs against LPSs, the mean OD value for Ross 308 broilers was 0.749 ± 0.390, while for Sasso broilers it was 1.636 ± 0.669, indicating a 1.9-fold higher level (*p* < 0.0001).

The differences in IgM anti-LPS NAb levels between the two groups were more pronounced than those for IgM anti-actin NAbs, underlining the importance of the former. Based on this finding, we further analyzed IgM anti-LPS measurements to assess their potential utility in differentiating free-range Sasso from conventional Ross 308 broilers through multiple logistic regression analysis (Figure 2A). For anti-LPS IgM NAb levels and a 95% confidence interval, the mean corresponds to 73% sensitivity and 83% specificity (Appendix A). The area under the ROC curve (AUC) for predicting conventional Ross 308 broilers was 0.829 (0.779–0.879, *p* < 0.0001). Additionally, Figure 2B shows the percentage of frequency distribution (*x*-axis) and the number of individual values (*y*-axis) to visualize the differences in data distribution between the FR and C groups.

### 3.2. Biochemical Parameters

The biochemical characteristics of broilers are shown in Table 2. Serum total cholesterol, triglycerides, creatinine, and ALT activity levels were similar between the two groups. Furthermore, no significant difference in 8-isoprostane levels, a biomarker for lipid peroxidation and oxidative stress, was observed between the two groups. The FR group exhibited significantly lower levels of serum AST and CPK activity compared to the C group (*p* < 0.001). Serum CPK levels were estimated to be about 3.7-fold higher in the C group compared to the FR group. Specifically, as shown in Figure 2, serum CPK activity is 11.6 ± 7.6 U/mL in the C group, while serum CPK activity in the FR group is 3.51 ± 2.6 U/mL (*p* < 0.001). Interestingly, the total serum of CPK activity in the C group is mainly distributed in the skeletal muscle isoform (CPK-MM) (Table 2 and Figure 3), while the total serum CPK activity in the FR group is distributed between CPK-MM and cardiac muscle isoform (CPK-MB), with a higher proportion in the CPK-MM isoform (Figure 4). The CPK-MM isoform levels were 20% higher in the C group compared to the FR group. In particular, the percentage (%) of CPK-MM isoform activity in the C group was 95.4 ± 4.2% versus 75.7 ± 7.2% in the FR group (*p* < 0.001).

As the FR group exhibited significantly lower serum AST and CPK activity compared to the C group (*p* < 0.001), with serum CPK levels approximately 3.7-times higher in the C group, we further evaluated CPK’s potential utility in differentiating free-range Sasso from conventional Ross 308 broilers by multiple logistic regression analysis (Figure 5A). In the results, for a 95% confidence interval, the mean corresponds to 94% sensitivity and 83% specificity (Appendix A). The area under the ROC curve (AUC) of CPK for predicting discrimination of conventional Ross 308 broilers was 0.888 (0.851–0.925, *p* < 0.0001). Additionally, Figure 5B shows the percentage of frequency distribution (*x*-axis) and the number of individual values (*y*-axis) to visualize the differences in data distribution between the FR and C groups.

### 3.3. Combined Use of Immunological and Biochemical Poultry Biomarkers

The above results show that IgM anti-LPS NAb levels, as an immunological parameter, and CPK levels, as a biochemical parameter, meet the criteria of potential biomarkers, and, therefore, we proceeded to combined multiple logistic regression analysis. The results of the analysis are shown in Figure 6. From the ROC curve for a 95% confidence interval, the mean corresponds to 84% sensitivity and 90% specificity. Additionally, unlike the previous single parameter analysis, this model was proven to be correct according to the null hypothesis (Appendix A), increasing the power of the analysis. The area under the ROC curve (AUC) of CPK for predicting the discrimination of conventional broilers was 0.955 (0.935–0.975, *p* < 0.0001).

## 4. Discussion

Blood biomarkers offer a clear and objective way to distinguish between free-range and conventionally raised broilers, helping to assess animal health and welfare, and optimize productivity [22,23]. For breeders and producers, these markers can validate free-range labeling, boost market differentiation, and support higher welfare practices that may command premium prices [24]. The findings presented herein have practical implications for improving poultry health monitoring, refining rearing practices, and enhancing product marketing [25,26].

Understanding the interplay among rearing systems, immunological parameters, and biochemical factors is crucial for optimizing poultry production. Previous studies have highlighted different immunological and biochemical parameters, such as antibiotic treatment, gut microbiome structure, and various metabolites, of broilers that profoundly deviate from poultry-rearing systems [27]. For example, genetic parameters of auto-antigen binding IgM and IgG antibodies in healthy chickens can serve as informative parameters for disease resistance, immune regulation, and maintenance of homeostasis [28]. Innate immunity plays an important role in the survival of organisms, since low levels of innate immunity may be associated with increased susceptibility to disease [29]. Measurements of IgM and IgY NAb levels have previously been reported to be suitable for poultry breed selection [30], as high serum levels of anti-KLH NAbs have been associated with high specific antibody responses and chicken survival [31]. Moreover, the results from these studies confirm that, while NAbs are partially heritable, maternal effects should also be considered [32]. Although anti-actin and anti-LPS NAb levels have been previously analyzed in the poultry sector, the analysis focused on total NAbs (of all Ig classes together) or IgY NAbs, indicating that their levels can be transiently increased by exogenous stimuli and therefore be susceptible to environmental or dietary stimuli [33,34]. In addition, in our previous study conducted under 3 years of industrial-scale production, we highlighted the importance of IgM versus IgY NAb levels’ measurements in relation to genotype and rearing systems; IgM NAb levels were shown to be influenced by genotype, rearing systems, and diet [20]. Herein, along with biochemical parameters, we analyzed anti-actin and anti-LPS IgM NAb levels as the best candidates for their potential application as biomarkers in the poultry industry.

On the other hand, CPK serves as a pivotal enzyme in the energy metabolism of tissues with significant in vivo energy fluctuations, particularly in skeletal muscle tissue, which has paramount importance in the meat industry [35]. Increased CPK activity has been associated with muscle fiber damage, internal degradation of skeletal muscle and connective tissue, inflammatory response, macrophage infiltration, etc., in studies performing intense and prolonged exercises [36]. CPK is also released in the blood after pain symptoms, motor limitation, and muscle spasms. In addition, factors, such as age, sex, physical condition, and season, are associated with increased variations in this enzyme [37,38]. Accumulating studies show that CPK activity can be used to assess muscle cell damage [39]. Stress and muscle damage alter the permeability of muscle cell membranes, resulting in increased levels of certain CPK isoforms [40]. Moreover, CPK is important for meat quality in animal production, especially in poultry production and growth [41]. Measurement of serum CPK levels has been used to assess sensitivity to physiological stress in systemic production in pigs, cattle, goats, red deer, and poultry [42,43,44,45]. CPK has also been used as an indicator of pre-slaughter physical stress and muscle damage, with levels directly related to animal welfare and meat quality [46,47,48,49,50]. Although many studies have shown that serum CPK activity is an important biochemical indicator in the various stages of industrial meat production, it remains unclear whether CPK activity can be used as a specific biomarker in poultry production. Based on our experimental data, the present study reveals significant variations in CPK levels between the conventional and free-range groups, with notably lower serum CPK concentrations observed in the free-range Sasso broilers. This significant difference appears to be due to the different breed, but also to the reduced mobility in conventional rearing conditions [51]. Although the growth and rearing protocols are closely followed, marked changes in movement can cause injury, damage, or spasms reflected in the secretion of CPK and other proteins and potential substances through the muscle membrane, confirmed by increased plasma AST activity in the conventional group. In addition, a positive correlation was observed between CPK and AST in the groups. Our data demonstrate that the distribution of total serum CPK activity primarily consists of the CPK-MM isoform, which accounts for over 70% of circulating CPK in the study groups. Notably, the CPK-MM levels in the conventional group were significantly higher than those in the free-range group. These findings suggest that CPK is a valuable biomarker for differentiating conventional and free-range broilers.

The results of this study highlight the applicability of the combined use of anti-LPS IgM NAbs and CPK levels as a novel method to distinguish fast-growing conventional Ross 308 from slow-growing Sasso free-range broilers. More specifically, we demonstrated that circulating anti-LPS IgM NAb levels and serum CPK activity levels effectively discriminate these poultry genotypes in our production model. These findings may have further applications in meat quality control, assessment of immune status, muscle cell damage following muscle strain or oxidative stress, animal productivity and poultry development, as well as biologically sustainable production. Overall, measurements of serum anti-LPS IgM NAb and CPK levels can be used to assess sustainability criteria and quality assurance requirements for poultry production from breeding farms, hatcheries, and production farms to factory processing and final shipment.

This study offers promising insights into the use of biomarkers for distinguishing between broiler-rearing systems. However, further research involving broader sample sizes and varied locations may help strengthen the applicability of these findings across different environmental and management contexts. Exploring a wider variety of stress, inflammation, and metabolic biomarkers could further expand the understanding of physiological differences between systems. In addition, future studies considering approaches that capture changes over time could provide a more dynamic view of biomarker patterns. Overall, by understanding the differences in immune function and tissue health between systems, breeders can adopt strategies that reduce disease risk and lower antibiotic use, promoting further both animal welfare and productivity.

## 5. Conclusion

This study introduces anti-LPS IgM natural antibodies and creatine phosphokinase as potent biomarkers to differentiate free-range from conventionally raised broilers, which is a novel and important contribution to poultry science. Their combination achieved high sensitivity (90%) and specificity (84%) in distinguishing between free-range Sasso and conventional Ross 308 broilers, making it a promising, cost-effective, and easy-to-use, diagnostic tool. These potent biomarkers provide a comprehensive assessment of physiological differences between different rearing systems, offering insights that could be beneficial for animal welfare and production optimization.

## Figures and Tables

**Figure 1 animals-14-03070-f001:**
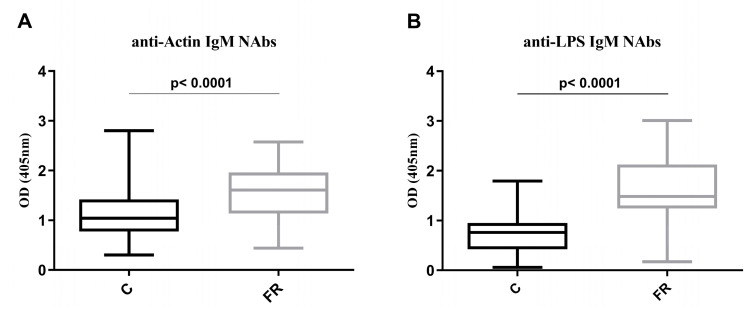
Levels of serum anti-actin and anti-LPS IgM NAbs. The *x*-axis shows conventional (C) Ross 308 (*n* = 150) and free-range (FR) Sasso (*n* = 150) broilers, while the *y*-axis displays the optical density (OD) for: (**A**) anti-actin IgM NAb levels and (**B**) anti-LPS IgM NAb levels. Statistical differences were assessed by *t*-test, with the significance level set at 5%.

**Figure 2 animals-14-03070-f002:**
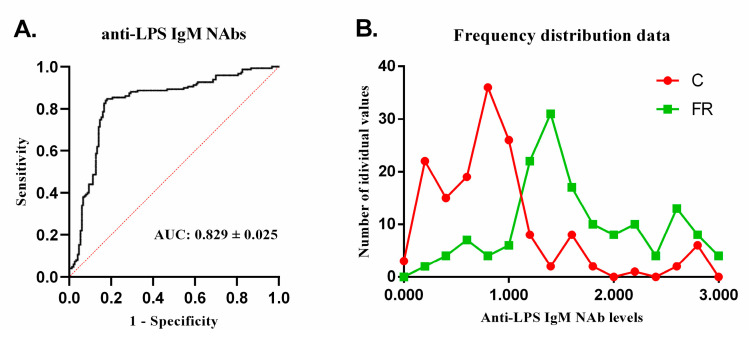
Anti-LPS IgM NAbs multivariate logistics analysis. (**A**) ROC curve analysis to detect the discriminative limit of anti-LPS IgM NAb levels between the two distributions of conventional and free-range broilers. (**B**) Frequency distribution of IgM anti-LPS data and number of individual values for conventional (C) Ross 308 (*n* = 150) and free-range (FR) Sasso broilers (*n* = 150).

**Figure 3 animals-14-03070-f003:**
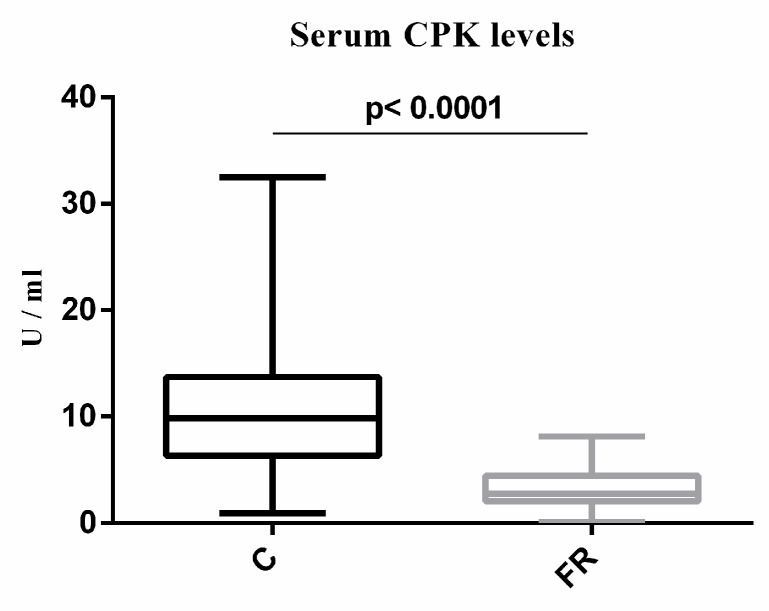
Levels of serum CPK activity. The *x*-axis shows conventional (C) Ross 308 (*n* = 150) and free-range (FR) Sasso broilers (*n* = 150), while the *y*-axis displays CPK activity measured in U/mL. Statistical differences were assessed by *t*-test, with the significance level set at 5%.

**Figure 4 animals-14-03070-f004:**
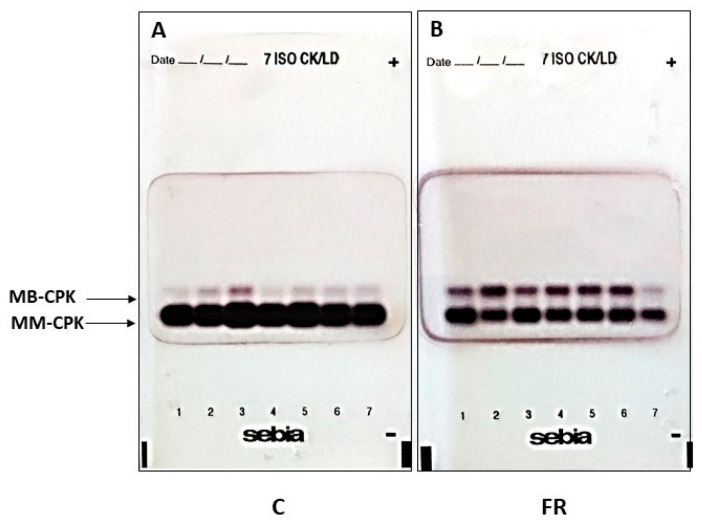
Representative profile of serum CPK isoforms of MM (skeletal muscle) and MB (cardiac muscle) types separated by agarose gel electrophoresis. (**A**) CPK isoforms of conventional (C) Ross 308 broilers and (**B**) CPK isoforms of free-range (FR) Sasso broilers. The kits are used in conjunction with the HYDRASYS semi-automated electrophoresis system. Separated CPK isoforms are visualized using a specific chromogenic substrate.

**Figure 5 animals-14-03070-f005:**
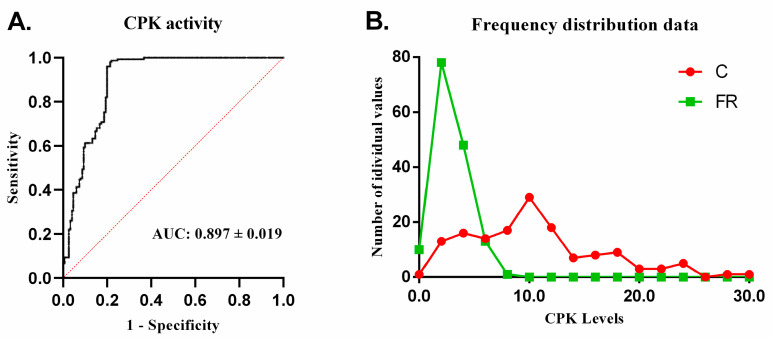
CPK multivariate logistics analysis (**A**). ROC curve analysis detecting the discriminative threshold of CPK enzyme activity between the two distributions of conventional and free-range broilers. (**B**). Frequency distribution of CPK data and number of individual values for conventional (C) Ross 308 (*n* = 150) and free-range (FR) Sasso broilers (*n* = 150).

**Figure 6 animals-14-03070-f006:**
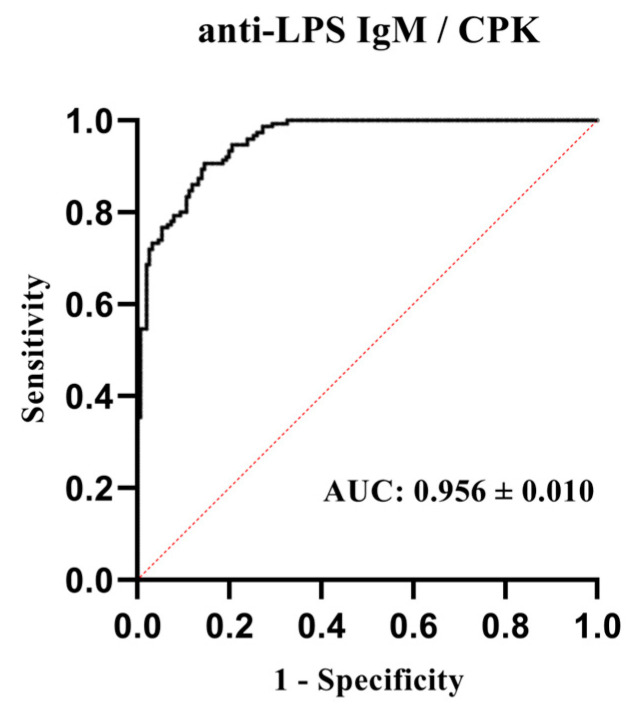
Combined use of the immunological and biochemical biomarkers. ROC curve analysis determining the discriminative limit of anti-LPS IgM NAbs and CPK levels between conventional Ross 308 (*n* = 150) and free-range Sasso (*n* = 150) broiler groups.

**Table 1 animals-14-03070-t001:** Characteristics of broilers and specifications of rearing systems.

	Conventional (C) (*n* = 150)	Free-Range (FR) (*n* = 150)
Genotype	Ross 308	Sasso
Growth rate	Fast-growing	Slow-growing
Slaughter/sampling	Day 47	Day 67
Housing	15 birds/m^2^	13 birds/m^2^ indoors and 1 bird/m^2^ in forage paddock
Pasture	Indoors	Grass
Starter	Day 1–17 (9 L:15 D light scheme)	
Grower	Day 18–35 (light > 10 min/day)	
Finisher	Day 36–slaughter (16 L:8 D light scheme)	

**Table 2 animals-14-03070-t002:** Results on biochemical parameters of the analysis.

Parameters	C Group	FR Group	*p*-Value
Cholesterol (mg/dL)	100.8 ± 17.4	101.5 ± 11.9	NS
Triglycerides (mg/dL)	42.3 ± 10.4	45.4 ± 11.9	NS
Creatinine (mg/dL)	0.22 ± 0.09	0.28 ± 0.10	NS
8-Isoprostane (ng/mL)	2.5 ± 1.6	2.9 ± 3.1	NS
ALT (U/L)	1.28 ± 0.64	1.69 ± 0.71	NS
AST (U/L)	50.36 ± 24.6 *	27.3 ± 10.9 *	<0.001
CPK (U/mL)	11.6 ± 7.6 *	3.51 ± 2.6 *	<0.001
CPK-ΜΜ (%)	95.4 ± 4.1 *	75.7 ± 7.2 *	<0.001

*: Statistically significant between groups; NS: not significant.

## Data Availability

The data presented in this study are available on request from the corresponding authors due to intellectual property restrictions.

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
