# Peer review of "Pasture vs. Coop: Biomarker Insights into Free-Range and Conventional Broilers"

_animals, 2024, doi:10.3390/ani14213070_

Round 1
Reviewer 1 Report
Comments and Suggestions for Authors
The study introduces IgM Natural Antibodies (NAbs) and creatine phosphokinase (CPK) as potential biomarkers to differentiate between free-range and conventionally raised broilers, which is a novel and important contribution to poultry science.
The combination of anti-LPS IgM-NAbs and CPK levels achieved high sensitivity (90%) and specificity (84%) in distinguishing between free-range Sasso broilers and conventional Ross 308 broilers, making it a promising diagnostic tool. By focusing on both immunological and tissue-related parameters, the study provides a comprehensive assessment of physiological differences between different rearing systems, offering insights that could be beneficial for animal welfare and production optimization.
The research findings have practical applications for the poultry industry, particularly in ensuring product differentiation, enhancing animal welfare practices, and improving marketing claims regarding free-range and conventional broilers.
Although the study used 300 broilers, larger and more geographically diverse sample sizes might be necessary to fully validate the biomarkers and ensure the results are widely applicable.
The study may have been conducted in a single location, which could limit the generalizability of the findings across different environmental conditions, rearing systems, or management practices.
While the selected biomarkers (IgM-NAbs and CPK) are promising, other potentially relevant biomarkers related to stress, inflammation, or metabolism were not explored. This might have provided a more robust understanding of the physiological differences between the systems.
The study does not provide longitudinal data showing how these biomarkers evolve, limiting the understanding of how rearing practices influence broiler health throughout the entire production cycle.
The ability to distinguish between free-range and conventionally raised broilers using biomarkers provides an objective method for assessing the health and welfare of animals, supporting the shift towards more ethical and sustainable rearing systems.
For breeders and producers, these biomarkers could be used to differentiate between products in the market, reinforcing free-range labeling claims and potentially commanding a premium for higher-welfare poultry products.
The study's findings offer breeders tools for monitoring bird health more precisely, enabling them to make more informed decisions on rearing practices that improve both animal welfare and productivity.
By understanding how immune function and tissue health differ between systems, breeders can adopt better strategies for disease prevention, ultimately reducing the reliance on antibiotics.
Future studies could include additional biomarkers related to stress, inflammation, and metabolic health to provide a more complete understanding of the physiological differences between free-range and conventional broilers.
To increase the generalizability of the findings, the study should be repeated with larger sample sizes and across multiple geographic locations and environmental conditions.
Conducting longitudinal studies to track biomarker levels throughout the birds' life cycle would help in understanding the long-term impacts of rearing systems on broiler health and welfare.
The authors could include an analysis of the economic feasibility of adopting these biomarkers in routine poultry management, providing breeders with a clearer picture of the costs and benefits.
Author Response
REVIEWER 1
Response to Comments and Suggestions for Authors: “The study introduces IgM Natural Antibodies (NAbs) and creatine phosphokinase (CPK) as …………… The research findings have practical applications for the poultry industry, particularly in ensuring product differentiation, enhancing animal welfare practices, and improving marketing claims regarding free-range and conventional broilers”.
We thank the reviewer for their review and thoughtful comments. We appreciate their overall positive evaluation and have carefully considered their suggestions to improve our manuscript. Please find bellow a point by point answer to reviewers’ comments. Lines are referring to revised manuscript with track changes.
Point-By-Point answers:
Comments 1: Although the study used 300 broilers, larger and more geographically diverse sample sizes might be necessary to fully validate the biomarkers and ensure the results are widely applicable. To increase the generalizability of the findings, the study should be repeated with larger sample sizes and across multiple geographic locations and environmental conditions.
Response 1: We have revised the manuscript to address this concern by adding a paragraph in the discussion section that highlights the limitations of the study (lines 427-436).
Comments 2: The study may have been conducted in a single location, which could limit the generalizability of the findings across different environmental conditions, rearing systems, or management practices.
Response 2: We appreciate the reviewer’s insight regarding this potential limitation. These biomarkers should be explored in various rearing systems and management practices in future studies to ensure broader applicability and expand our conclusions. We have revised the manuscript to address this concern by adding a paragraph in the discussion section that highlights the limitations of the study (lines 427-436).
Comments 3: While the selected biomarkers (IgM-NAbs and CPK) are promising, other potentially relevant biomarkers related to stress, inflammation, or metabolism were not explored. This might have provided a more robust understanding of the physiological differences between the systems. Future studies could include additional biomarkers related to stress, inflammation, and metabolic health to provide a more complete understanding of the physiological differences between free-range and conventional broilers.
Response 3: We thank the reviewer for this comment. Biochemical parameters such as isoprostanes and Aminotransferases are considered as stress/inflammatory biomarkers while triglycerides and cholesterol as metabolic biomarkers. Other biomarkers regarding antioxidant status have been previously investigated; we added this information in the manuscript (Introduction lines 92-96). Since our objective in this study was to investigate the applicability of novel and more promising biomarkers, we focused on NAbs and CPK since their combination led to significantly higher sensitivity and specificity. We added a paragraph pointing out the limitation of this study in discussion (lines 427-436).
Comments 4: The study does not provide longitudinal data showing how these biomarkers evolve, limiting the understanding of how rearing practices influence broiler health throughout the entire production cycle. Conducting longitudinal studies to track biomarker levels throughout the birds' life cycle would help in understanding the long-term impacts of rearing systems on broiler health and welfare.
Response 4: As this study was conducted within industrial-scale production systems, continuous blood collection or individual bird monitoring is challenging and could affect performance and productivity. It is in our future plans to conduct such evaluations and analyses in the new "Experimental housing Unit," currently under development. We revised our manuscript accordingly to address and highlight this limitation in discussion (lines 427-436).
Comments 5: The ability to distinguish between free-range and conventionally raised broilers using biomarkers provides an objective method for assessing the health and welfare of animals, supporting the shift towards more ethical and sustainable rearing systems.
For breeders and producers, these biomarkers could be used to differentiate between products in the market, reinforcing free-range labeling claims and potentially commanding a premium for higher-welfare poultry products.
The study's findings offer breeders tools for monitoring bird health more precisely, enabling them to make more informed decisions on rearing practices that improve both animal welfare and productivity.
By understanding how immune function and tissue health differ between systems, breeders can adopt better strategies for disease prevention, ultimately reducing the reliance on antibiotics.
Response 5: We appreciate the reviewer’s valuable suggestions and insights, which have been carefully considered to enhance our manuscript, and are incorporated into the revised sections in discussion (lines 353-359).
Comments 6: The authors could include an analysis of the economic feasibility of adopting these biomarkers in routine poultry management, providing breeders with a clearer picture of the costs and benefits.
Response 6: We thank the reviewer for this thoughtful suggestion. We acknowledge that analysis of economic data is beyond the objectives of this study. However, the methods used, such as ELISA and spectrophotometry, are both cost-effective and straightforward for standard laboratory applications. This information has been added to the manuscript to further highlight their feasibility for practical use (Abstract lines 45-46 and conclusion lines 443-444)

Reviewer 2 Report
Comments and Suggestions for Authors
Pasture vs. Coop: Biomarker insights into Free-Range and Conventional Broilers.
This study aimed to evaluate the individual and combined utility of specific blood parameters as biomarkers to differentiate free-range, slow-growing Sasso broilers from conventionally raised fast-growing Ross 308 broilers.
Our results showed significantly higher IgM-NAb levels to both antigens and lower CPK and Aspartate Aminotransferase levels in Sasso broilers compared to Ross 308 broilers.
There are few points which will make the manuscript reads better.
The abstract needs revision, L26-28: remove, Add experimental design.
The introduction: revise, add paragraph about free range chicken, add more information about Sasso breed.
L72: Ross 308 is a strain of chicken, correct here and every where
L78-85: this part should be added to the materials and methods.
300 chickens from each group?
Tabl1 1: Light and dark hours based on what recommendation?
All tables and figures: add a description under all tables and figures, include description of all abbreviations. Add n "number of samples used for each analysis"
L312-314: mentioned earlier, revise the discussion.
L312-320: revise.
Add conclusion part to summarize the major results and applications.
Author Response
REVIEWER 2
Comments and Suggestions for Authors: “This study aimed to evaluate ……There are few points which will make the manuscript reads better”.
We thank the reviewer for their review and thoughtful comments. We appreciate their overall positive evaluation and have carefully considered their suggestions to improve the manuscript. Please find bellow a point by point answer to reviewers’ comments. Lines are referring to revised manuscript with track changes.
Point-By-Point answers:
Comment 1: The abstract needs revision, L26-28: remove, Add experimental design.
Response 1: We revised accordingly in the Abstract section (lines 26-30 and 38-39).
Comment 2: The introduction: revise, add paragraph about free range chicken, add more information about Sasso breed.
Response 2: We thank the reviewer for this valuable comment. A new paragraph in the revised manuscript has been added in the Introduction (lines 69-82).
Comment 3: L72: Ross 308 is a strain of chicken, correct here and every where
Response 3: Thank you for this correction. We revised accordingly throughout the manuscript.
Comment 4: L78-85: this part should be added to the materials and methods.
Response 4: We revised accordingly and this paragraph is added in the materials and methods section (lines 126-131).
Comment 5: 300 chickens from each group?
Response 5: The broilers enrolled in this study were 300 in total (150/group). We revised accordingly by adding this information to the main text in the Abstract (line 38) and materials and methods (lines 115-116).
Comment 6: Table1 1: Light and dark hours based on what recommendation?
Response 6: We revised accordingly by adding the appropriate references at the materials and methods section, lines 11-129, Ref:19.
Comment 7: All tables and figures: add a description under all tables and figures, include description of all abbreviations. Add n "number of samples used for each analysis"
Response 7: We revised accordingly throughout the manuscript by adding a description under all tables and figures as well as "n" number of samples used for each analysis.
Comment 8: L312-314: mentioned earlier, revise the discussion.
Response 8: We assumed that the reviewer refers to the first lines of the Discussion. We revised accordingly the first paragraph of the discussion (lines 347-359).
Comments 9: L312-320: revise.
Response 9: We revised accordingly the first paragraph of the discussion (lines 347-359).
Comments 10: Add conclusion part to summarize the major results and applications.
Response 10: We thank the reviewer for helping us improve our manuscript. We revised accordingly by adding conclusion section lines 438-446.
